# Atmospheric vertical structure variations during severe aerosol pollution events based on lidar observations

Qimeng Li<sup>1,2</sup>, Huige Di<sup>1</sup>, Ning Chen<sup>1</sup>, Xiao Cheng<sup>1</sup>, Jiaying Yang<sup>1</sup>, Yun Yuan<sup>1</sup>, Qing Yan<sup>1</sup>, and Dengxin Hua<sup>1</sup>

<sup>1</sup>School of Mechanical and Precision Instrument Engineering, Xi'an University of Technology, Xi'an 710048, China

Correspondence: Huige Di (dihuige@xaut.edu.cn)

Abstract. During severe haze events, the boundary layer exhibits a complex vertical structure, while high aerosol loadings hinder high-resolution temperature and humidity measurements. To address this, a Raman-Mie lidar and retrieval algorithms for temperature, humidity, and aerosol optical properties were developed at Xi'an University of Technology, enabling high-resolution profiling of haze vertical structures. A 12-day haze episode was continuously monitored from formation to dissipation, providing detailed spatiotemporal variations of temperature, relative humidity, and aerosols. The boundaries of temperature inversion (TI) and aerosol layers were identified using a threshold method. The results revealed a strong coupling between aerosols and temperature during pollution evolution. Dome and stove effects were observed, with possible coexistence and interaction. Three dome-shaped TIs were identified. The top of a decreasing-type aerosol layer formed a stratified dome structure that constrained vertical diffusion, with the temperature gradient of the elevated TI varying inversely with its depth. Both TI strength and humidity were strongly correlated with surface  $PM_{2.5}$  concentrations. Surface-based TI exhibited a clear diurnal variation, with TI peaks preceding aerosol peaks. The results indicated that strong elevated TI and weak turbulence in the lower layer favored aerosol accumulation. Clouds and virga not only suppressed radiative heating but also enhanced humidity, further driving the rapid increase in surface  $PM_{2.5}$  concentrations. During the dissipation stage, the rapid breakdown of TI and enhanced solar heating were critical for pollutant removal, while efficient horizontal transport facilitated the complete clearance of aerosols within the boundary layer.

#### 1 Introduction

Air pollution control has long been a key environmental issue accompanying rapid economic development and remains closely linked to public health (Fu et al., 2014; Wang et al., 2023; Song et al., 2017; Di et al., 2021). Aerosols, consisting of fine solid particles and liquid droplets suspended in the atmosphere (Di et al., 2021), represent a key medium for exploring pollution formation and evolution processes. Although excessive aerosol emissions resulting from industrial activities and intensive human socioeconomic processes are the fundamental drivers of frequent pollution events (Zhang et al., 2012, 2013; Le et al., 2020; Wu et al., 2025), meteorological conditions often act as the dominant triggers and regulators of these events (Su et al., 2004; Zhang et al., 2016; Zhu et al., 2018; Zhong et al., 2019; Huang et al., 2021; Jiang et al., 2021; Prasad et al., 2022).

<sup>&</sup>lt;sup>2</sup>School of Electronic Information, Shaanxi Institute of Technology, Xi'an 710300, China

50

There exists a strong two-way feedback between the vertical distribution of aerosols and meteorological factors (Zhong et al., 2018, 2019). Meteorological conditions unfavourable for aerosol dispersion, such as TI, weak winds, and high relative humidity, are generally considered typical features that exacerbate pollution severity (Zhong et al., 2018, 2019; Miao et al., 2019; Liu et al., 2022; Shao et al., 2023; Zhou et al., 2024). Temperature inversion (TI), characterized by an increase in temperature with height, directly alters atmospheric buoyancy and suppresses vertical mixing (Su et al., 2020). Weak winds restrict horizontal dispersion of aerosols, further facilitating their accumulation (Sekuła et al., 2021). Meanwhile, the hygroscopic growth of aerosols intensifies with increasing relative humidity, serving as a major driver of rapid pollution deterioration (Zhong et al., 2018, 2019). These factors interact synergistically, jointly contributing to the formation and development of surface haze pollution. Huang et al. (2021) investigated the vertical thermal structures under polluted conditions and reported pronounced heating above the planetary boundary layer (PBL), accompanied by cooling near the surface. This pattern, known as the aerosol dome effect (Ding et al., 2016), enhances surface pollutant accumulation by stabilizing the lower atmosphere. Conversely, when aerosols are concentrated near the surface, their absorption of solar radiation heats the lower layer, increases turbulent mixing, and partially alleviates surface pollution—a process termed the aerosol stove effect (Wang et al., 2018). Although numerous studies have explored the interactions between aerosols and meteorological conditions during pollution episodes, most rely on discontinuous or large-scale inferential analyses rather than direct high-resolution observations. Continuous, high spatiotemporal-resolution measurements of boundary-layer structures—particularly temperature and humidity profiles—remain scarce due to observational limitations. This lack of detailed vertical observations hampers a comprehensive understanding of the fine-scale evolution of boundary-layer thermodynamic structures and their coupling with aerosol processes during haze development.

Lidar, characterized by high accuracy and high spatiotemporal resolution, serves as an effective tool for obtaining vertical profiles of temperature, humidity, and aerosols, thereby enabling refined investigations of small-scale atmospheric processes and extreme weather phenomena (Wulfmeyer et al., 2015; Girolamo et al., 2016; Lange et al., 2019; Yi et al., 2021; Li et al., 2022, 2023, 2025). Under clear-sky conditions, lidar can provide highly precise temperature and humidity measurements. However, under high aerosol loading (i.e., during haze pollution), signal crosstalk may induce substantial deviations in retrieved temperature profiles (Su et al., 2013; Li et al., 2022, 2023, 2025). This issue mainly arises because strong Mie scattering signals contaminate the rotational Raman channel used for temperature detection, leading to retrieval errors. In addition, discrepancies in the geometric overlap factors among Raman channels can introduce systematic biases in temperature and humidity retrievals. These limitations make accurate thermodynamic profiling particularly challenging in polluted atmospheres. To address these challenges, Xi'an University of Technology (XUT) developed a multi-wavelength Raman-Mie scattering lidar system. Di, Li, and co-workers conducted comprehensive studies on temperature and humidity retrieval techniques under hazy and cloudy conditions and proposed correction algorithms for rotational Raman temperature retrievals and geometric overlap effects. These advancements enabled high-precision retrievals of temperature, humidity, and aerosol profiles within clouds and throughout the atmospheric boundary layer (Li et al., 2022, 2023, 2025).

In the winter of 2023, a severe haze pollution episode was continuously observed using the XUT Raman-Mie scattering lidar system. By applying the aforementioned retrieval algorithms, high-accuracy profiles of temperature, humidity, and

https://doi.org/10.5194/egusphere-2025-5393 Preprint. Discussion started: 27 November 2025

aerosol optical properties within the haze layer were obtained, allowing detailed characterization of the vertical evolution of the atmosphere during the formation, development, and dissipation stages of the event. Multiple auxiliary datasets, including radiosonde soundings and ERA5 reanalysis data, were integrated to examine the coupling between meteorological conditions and pollution evolution. The observations revealed a distinct dome-shaped stratification of aerosols induced by solar radiation, demonstrating the coexistence of both the dome effect and the stove effect. Furthermore, the temporal evolution of temperature inversions (TIs) and their regulatory influence on the vertical distribution of aerosols were clearly captured throughout the pollution episode. Correlation analyses were conducted between surface PM<sub>2.5</sub> concentrations and thermodynamic variables, such as temperature and relative humidity, alongside assessments of the diurnal variations in temperature gradients.

# 2 Data, observation locations, and methods

### 2.1 Data

The primary datasets employed in this study include lidar-derived profiles of temperature, humidity, and aerosol optical properties; radiosonde soundings; surface meteorological observations; surface particulate matter concentrations ( $PM_{10}$  and  $PM_{2.5}$ ) obtained from air quality monitoring stations; surface radiation measurements; and vertical atmospheric profiles from the ERA5 reanalysis dataset.

The Raman-Mie scattering lidar employed in this study was developed by Xi'an University of Technology in 2021. The system uses a 354.7 nm Nd: YAG laser as the excitation source and a 400 mm-aperture Cassegrain telescope to collect atmospheric backscatter signals. It provides a minimum vertical resolution of 3.75 m and a temporal resolution of 2 min. Atmospheric temperature profiles are retrieved using a rotational Raman method, whereas water vapour profiles are derived via a combined vibrational-rotational Raman approach. A vibrational Raman channel for nitrogen further enables accurate retrievals of aerosol extinction and backscatter coefficients using the Raman technique, with retrieval uncertainties below 15 %. For temperature retrievals, a 30 min temporal accumulation and 60-point smoothing are applied to enhance the signal-to-noise ratio, while water vapour retrievals use a 10 min accumulation and 30-point smoothing. Relative humidity is calculated from the lidar-derived temperature and water vapour profiles. Comparisons with co-located radiosonde measurements show standard deviations of less than 1 K for temperature and 0.5 g kg<sup>-1</sup> for water vapour. The main technical specifications of the lidar system are summarised in Table 1.

Radiosonde data were obtained from the Xi'an Jinghe Meteorological Basic Station, where soundings were launched daily at 07:15 and 19:15 Beijing Time. The radiosonde model used was GTS13, transmitting measurements at a frequency of 1 Hz, with a vertical resolution of 6 m in the lower atmosphere and a maximum detection height of 30 km. When the ambient temperature exceeds -80 °C, the temperature measurement accuracy is  $\pm 0.2$  °C. The relative humidity measurement error is less than 6% under ambient temperatures above -20 °C, and for atmospheric pressures above 500 hPa, the pressure measurement error is below 1.5 hPa. The instrument also provides horizontal wind speed and direction.

Table 1. Main technical specifications

| Specification              | Main parameters                              |                                  |                                                    |
|----------------------------|----------------------------------------------|----------------------------------|----------------------------------------------------|
| Receiving module           | Cassegrain telescope                         |                                  | HFOV: 0.5 mrad; TA: 400 mm                         |
| Transmitting module        | Nd: YAG Laser                                |                                  | EW:354.7nm; EPP: ~220mJ; PD:9ns                    |
| Spectral separation system | High quantum number rotational Raman channel |                                  | FWHM: 0.9 nm @352.5 nm, ESSR~65dB @354.7 nm        |
|                            | Low quantum number rotational Raman channel  |                                  | FWHM: 0.5 nm @353.9 nm, ESSR~50dB @354.7 nm        |
|                            | Water vapor vibrational Raman channel        |                                  | FWHM: 0.5 nm @407.5 nm, ESSR~75dB @354.7 nm        |
|                            | Nitrogen vibrational Raman channel           |                                  | FWHM: 0.5 nm @386.7 nm, ESSR $\sim$ 75dB @354.7 nm |
|                            | Mie-Rayleigh Channels                        |                                  | FWHM: 0.5 nm @354.7 nm                             |
|                            |                                              |                                  | FWHM: 0.5 nm @1064 nm                              |
| Signal                     | Photo multiplier tube (PMT)                  |                                  | Quantumefficiency:0.26, Hamamatsu                  |
| acquisition                | Multi-channel data acquisition card          |                                  |                                                    |
| Retrieval algorithm        | Temperature                                  | Rotational Raman method          | Deviation of temperature: 1 K                      |
|                            |                                              | Raman ratio correction technique |                                                    |
|                            | Water vapor                                  | Vibrational Raman method         | Deviation of water vapor: <0.5 g kg <sup>-1</sup>  |
|                            | Aerosol                                      | Vibrational Raman method         | Deviation of backscatter coefficient: 15%          |

HFOV, half field of view; TA, telescope aperture; EW, excitation wavelength; EPP, energy per pulse; PD, pulse duration; FWHM, full width at half maximum; ESSR, elastic scattering suppression ratio.

Surface observations were collected from ground-based meteorological stations and particulate matter (PM) monitors, providing hourly measurements of near-surface temperature, humidity, atmospheric pressure, visibility, and PM concentrations. These surface measurements were co-located with the radiosonde launch site to ensure data consistency.

ERA5 is the fifth-generation global atmospheric reanalysis dataset produced by the European Centre for Medium-Range Weather Forecasts (ECMWF). It provides hourly estimates of a wide range of atmospheric, land, and oceanic variables on a regular latitude-longitude grid with a horizontal resolution of approximately 31 km (0.25° × 0.25°) and 137 model levels from the surface up to 0.01 hPa. ERA5 employs a four-dimensional variational data assimilation system to combine model output with a large number of quality-controlled observations, generating a physically consistent, spatially complete, and temporally continuous representation of the Earth system. The dataset includes variables such as temperature, humidity, horizontal and vertical winds, surface pressure, precipitation, and radiation, available at the surface and multiple pressure or model levels. ERA5 data were accessed from the Copernicus Climate Data Store (CDS, https://cds.climate.copernicus.eu). In this study, we focused on key meteorological parameters, including isobaric maps, horizontal wind vectors, and vertical motion, which were used to characterize the background atmospheric conditions.

125

#### 2.2 Observation locations

The above instruments were located at the Jinghe National Meteorological Station in Xi'an, Shaanxi Province, China (34°26'N, 108°58'E), situated on the north bank of the Wei River in the north-central region of the Guanzhong Plain (33°42'-34°45'N, 107°40'-109°49'E, about 400 m above sea level). The Guanzhong Plain is a typical elongated basin-oriented east-west, bordered by the Qinling Mountains to the south and the northern Loess Plateau to the north. Elevations are higher along the southern and northern margins and gradually decrease toward the east. This enclosed or semi-enclosed topography restricts both horizontal and vertical air exchange, impeding the dispersion of atmospheric pollutants. In winter, the influence of the westerlies weakens, and the frequency and intensity of cold air outbreaks decrease. Under these conditions, stable stratification and TIs occur more frequently, and the boundary layer is typically lower, further limiting pollutant dilution and transport. Consequently, the Guanzhong Plain is prone to prolonged, large-scale, and severe air pollution episodes during winter.

#### 2.3 Lidar data retrieval algorithm

# 2.3.1 Retrieval algorithms for temperature, humidity, and backscatter coefficient within haze layers

The fundamental principle for probing atmospheric vertical structures using a Raman-Mie scattering lidar is based on the lidar equation, which involves the theories of Mie-Rayleigh scattering, rotational Raman scattering, and vibrational Raman scattering. This can be represented by Eq.(1)-(3). Specifically, rotational Raman scattering is used for temperature measurements, while Mie-Rayleigh scattering and vibrational Raman scattering are employed for retrieving aerosol optical properties and water vapor profiles.

$$P_{\rm r}(T, z, \lambda_r) = \frac{C_{\rm r}}{z^2} \beta_{\rm r}(T, z, \lambda_r) \exp \left\{ -\int_0^z \left[ \alpha(z', \lambda_{\rm e}) + \alpha(z', \lambda_{\rm r}) \right] \mathrm{d}z' \right\}$$
 (1)

$$P_{\mathbf{v}}(z,\lambda_{\mathbf{v}}) = \frac{C_{\mathbf{v}}}{z^{2}} \beta_{\mathbf{v}}(z,\lambda_{\mathbf{v}}) \exp \left\{ -\int_{0}^{z} \left[ \alpha(z',\lambda_{\mathbf{e}}) + \alpha(z',\lambda_{\mathbf{v}}) \right] dz' \right\}$$
(2)

$$P_{\rm e}(z,\lambda_{\rm e}) = \frac{C_{\rm e}}{z^2} \left(\beta_{\rm a}(z,\lambda_{\rm e}) + \beta_{\rm m}(z,\lambda_{\rm e})\right) \exp\left\{-2\int_0^z \left[\alpha(z',\lambda_{\rm e})\right] \mathrm{d}z'\right\}$$
(3)

where  $P_{\rm r}$ ,  $P_{\rm v}$ , and  $P_{\rm e}$  denote the rotational Raman, vibrational Raman, and elastic backscattering signals, respectively. C represents the system constant of the lidar, which includes the laser power, telescope receiving area, optical transmittance, and detector quantum efficiency.  $\lambda$ , T, and z refer to the wavelength, temperature, and height, respectively, while  $\alpha$  and  $\beta$  denote the volume extinction and backscattering coefficients. The subscripts e, rh, rl, vw, and vn correspond to the Mie-Rayleigh signal, high-quantum-number rotational Raman signal, low-quantum-number rotational Raman signal, water vapor vibrational Raman




signal, and nitrogen vibrational Raman signal, respectively, whereas m and a represent the molecular and aerosol components. The atmospheric extinction coefficient can be derived from the nitrogen vibrational Raman signal  $(P_{vn})$ , while the backscatter ratio (BR) is obtained in combination with the Mie-Rayleigh scattering signal. Finally, the aerosol backscatter coefficient can be retrieved, as expressed in Eq.(4)-(6).

$$\alpha_{\rm a}(z,\lambda_{\rm e}) = \frac{\mathrm{d}[\ln(\frac{N(z)}{z^2 P_{\rm vn}})]/\mathrm{d}z - \alpha_{\rm m}(z,\lambda_{\rm e}) - \alpha_{\rm m}(z,\lambda_{\rm vn})}{1 + (\lambda_{\rm e}/\lambda_{\rm vn})^K} \tag{4}$$

$$Br(z, \lambda_{e}) = F \frac{C_{vn}}{C_{e}} \frac{P_{e}(z, \lambda_{e})}{P_{vn}(z, \lambda_{vn})} \exp \left\{ \int_{0}^{z} [\alpha(z', \lambda_{e}) - \alpha(z', \lambda_{vn})] dz' \right\}$$
(5)

$$\beta_{\rm a}(z,\lambda_{\rm e}) = Br(z,\lambda_{\rm e}) \cdot \beta_{\rm m}(z,\lambda_{\rm e}) - \beta_{\rm m}(z,\lambda_{\rm e}) \tag{6}$$

In Eq.(4), N(z) and K represent the atmospheric molecular number density and the Ångström exponent, respectively, which is typically taken as 1. In Eq.(5), F denotes the system constant used for calculating BR. The rotational Raman temperature retrieval is based on the correlation between the rotational Raman signal and atmospheric temperature. By extracting two rotational Raman backscattering signals with different temperature sensitivities, a functional relationship between their signal ratio and temperature can be established to derive the atmospheric temperature profile. However, the rotational Raman channels are highly susceptible to elastic scattering cross-talk, which may introduce biases in the temperature retrieval. In Eq.(7), A and B are system constants used for temperature retrieval.  $M_{\rm h}$  and  $M_{\rm l}$  represent the elastic scattering cross-talk coefficients for the high- and low-quantum-number rotational Raman channels, respectively. The water vapor and nitrogen vibrational Raman scattering signals are extracted, and their ratio is calculated. Based on radiosonde data obtained under the same temporal and spatial conditions, a regression relationship for data retrieval is constructed, enabling the lidar measurement of the water vapor mixing ratio and, consequently, the derivation of relative humidity, as expressed in Eq.(8). The aforementioned measurement principles are well established, and detailed descriptions can be found in previous studies (Ansmann et al., 1992; Li et al., 2023, 2025; Chazette et al., 2025).

$$T(z) = A / \left\{ \ln \left[ \frac{C_{\rm rl}(z)}{C_{\rm rh}(z)} \frac{P_{\rm rh}(T, z, \lambda_{\rm rh}) + M_{\rm h} \cdot P_{\rm e}(z, \lambda_{\rm e})}{P_{\rm rl}(T, z, \lambda_{\rm rl}) + M_{\rm l} \cdot P_{\rm e}(z, \lambda_{\rm e})} \right] - B \right\}$$

$$(7)$$

$$W(z) = \frac{C_{\text{vn}}}{C_{\text{vw}}} \frac{P_{\text{vw}}(z, \lambda_{\text{vw}})}{P_{\text{vn}}(z, \lambda_{\text{vn}})} \exp \left\{ \int_{0}^{z} \left[ \alpha(z', \lambda_{\text{vw}}) - \alpha(z', \lambda_{\text{vn}}) \right] dz' \right\}$$
(8)

Under atmosphere conditions with cloud layers or high aerosol concentrations, due to the close spectral intervals, strong elastic scattering can cause elastic scattering cross-talk in the low-quantum-number rotational Raman channel, and even in the



high-quantum-number channel. This means that a portion of the elastic scattering signal is mixed into the rotational Raman channels, which leads to significant deviations in data retrieval, and in severe cases, preventing accurate temperature retrieval within the haze layer. Since haze layers are generally located at low height, the geometric overlap factors of each lidar channel are not identical, which also causes large uncertainties in temperature retrievals in the lower atmospheric region. Therefore, temperature measurements in haze layers mainly include three processes: geometric overlap factor correction, rotational Raman ratio correction, and temperature retrieval.

First, a higher-height detection region, where the geometric overlap factor has no influence, is selected, and is combined with radiosonde temperature profiles obtained under the same temporal and spatial conditions to perform system calibration for temperature retrieval. Then, under clear-sky and dry near-surface conditions, the theoretical rotational Raman ratio is derived from radiosonde data, and by comparing it with the measured Raman ratio, the ratio of geometric overlap factors is obtained, achieving geometric overlap factor correction. Subsequently, the theoretical Raman ratio under strong elastic scattering conditions is derived from radiosonde data, and the measured Raman ratio is used to calculate the elastic scattering cross-talk ratio. By constructing a linear relationship between the elastic scattering cross-talk ratio and BR, the rotational Raman ratio correction is achieved. Finally, the vertical atmospheric temperature profile is retrieved. Figure 1 shows the temperature profiles obtained using the above retrieval algorithm. The detailed algorithm and procedures can be found in the published literature (Li et al., 2025).

**Figure 1.** Aerosol backscatter coefficient (BC) and temperature retrievals. Panels (a), (c), and (f) show BC, while (b), (d), and (e) show temperature. The black solid line represents BC, and the blue dashed, red solid, and black dash-dotted lines correspond to the uncorrected temperature profile, the corrected temperature profile, and the radiosonde temperature profile, respectively.




# 2.3.2 Retrieval algorithms for vertical velocity and buoyancy acceleration

The vertical velocity used in this study was derived from the vertical pressure tendency provided by ERA5. Atmospheric static stability is governed by buoyancy, which in turn influences the vertical distribution of aerosols. In this study, the buoyancy acceleration was estimated using an approach based on the environmental virtual potential temperature. This method assumes small perturbations and linearization, making it suitable for weakly disturbed conditions. Under strong perturbations, however, it may underestimate the actual buoyancy. Nevertheless, the estimated values can reasonably capture the overall trend of changes in thermodynamic stability. Buoyancy is expressed as (Su et al., 2020)

175 
$$B = \frac{\mathrm{d}^2 z_0}{\mathrm{d}t^2} = -g\Delta z \frac{1}{\theta} \frac{\mathrm{d}\theta}{\mathrm{d}z_0} \tag{9}$$

where  $z_0$  denotes the height of the air parcel, and t represents time.  $\theta$  is the virtual potential temperature of the environment, which is calculated using radiosonde pressure combined with the lidar-measured atmospheric temperature and water vapor mixing ratio. A positive buoyancy indicates a convective atmospheric layer, whereas a negative buoyancy corresponds to a stable layer. When the buoyancy approaches zero, the atmospheric column is neutral.

## 80 2.3.3 Method for Determining Aerosol Boundary and TI Characteristic

The aerosol layer boundary was identified using a gradient-based method applied to BR. The fundamental principle of this method is to locate positions exhibiting pronounced discontinuities in the BR profile by calculating its vertical derivative. As this study primarily focuses on the relationship between the aerosol vertical structure and temperature distribution, only the upper boundary of the aerosol layer (hereafter referred to as UABL) was determined, without further distinction of the stable boundary layer, mixed layer, and residual layer.

First, the first derivative of the BR profile (i.e., BR gradient) was calculated, and the negative-gradient intervals of BR (hereafter referred to as NIBRs) were identified based on the zero-crossing points of the derivative. To eliminate local perturbations and high-frequency noise, threshold conditions (> 60 m) were applied to both the NIBR lengths and the spacing between adjacent NIBRs, ensuring that only valid NIBRs were retained. Similarly, the positive-gradient intervals of BR (hereafter referred to as PIBRs) were identified through zero-crossing detection and filtered with the same threshold criteria. Second, gradient thresholds were defined for both PIBRs (> 3 km<sup>-1</sup>) and NIBRs (< -3 km<sup>-1</sup>). In addition, a minimum BR deviation threshold within each interval ( $|\Delta Br| > 0.35$ , defined as the absolute deviation between the maximum and minimum BR within the interval) was imposed to exclude pseudo-transition zones with weak gradients. When the length of a positive interval satisfying the gradient threshold exceeded 30 m, it indicated a notable enhancement of aerosol concentrations within that layer. In such cases, the adjacent negative interval immediately above was retained even if it did not meet the gradient threshold. Finally, the UABL was determined within NIBRs using a BR threshold of 1.3. Starting from the top of each NIBR, one-eighth of its length was used as the reference range. If the maximum BR within this range exceeded 1.3, the height corresponding to one-eighth of the NIBR length below the top was defined as the UABL. Otherwise, the height at which the BR first exceeded 1.3 within NIBRs was identified as the UABL.


**Figure 2.** Extraction of UABL and TI characteristics. Panels (a)-(d) show (a) BR, (b) BR gradient, (c) temperature, and (d) temperature gradient. In panels (a)-(d), the black curves represent the BR, BR gradient, temperature, and temperature gradient, respectively. Asterisks indicate the identified aerosol layer boundaries. In panel (b), the blue solid and red dashed shaded regions denote NIBR and PIBR, respectively. In panels (c) and (d), the red dashed shaded regions highlight the TI layers.

TI features were identified using a gradient-based method. Unlike conventional gradient methods, the raw temperature profile was first subjected to linear feature fitting prior to calculating the vertical temperature gradient. The fitting employed the piecewise linear interpolation method proposed by Fochesatto (2015), which constructs an interpolation function with variable segment lengths and minimizes an associated error function to obtain optimal fits for each segment, thereby accurately resolving TI layers.

The first derivative of the fitted temperature profile (i.e., the vertical temperature gradient) was then calculated, and the positive-gradient intervals of temperature (hereafter referred to as PITs) along with their adjacent gap layers were identified using zero-crossing points. Threshold conditions (> 60 m) were applied to both PIT lengths and gap layers to retain only valid segments. Gap layers shorter than the threshold were merged with adjacent PITs to form a single TI layer. Conversely, PITs shorter than the threshold, when adjacent to gap layers exceeding the threshold, were discarded as likely resulting from minor perturbations or weak TI features. To prevent underestimation of TI thickness, additional criteria were applied to isolated PITs. If a gap layer exceeded the threshold length and its mean vertical gradient exceeded -0.5 K km<sup>-1</sup>, or the maximum temperature deviation within the gap layer was less than 0.5 K, indicating nearly uniform temperature with height, the adjacent PITs were merged. PITs satisfying these conditions were thus considered valid TI layers. The base and top of each TI layer were defined as the TI base and TI top, respectively. The vertical distance between TI base and top represents TI thickness. The equivalent vertical temperature gradient was then calculated from the temperature difference within TI layer.





#### 3 Results and discussion

# 3.1 Overview of the haze pollution event

A prolonged haze pollution episode occurred in Xi'an in late December 2023, lasting for 12 days. Figure 3 illustrates the surface PM concentrations and meteorological parameters during this persistent pollution event. As shown in the figure, the surface aerosol concentrations began to accumulate on 22 December and continued to increase, reaching a peak in PM<sub>2.5</sub> concentrations on the evening of 28 December, which persisted until noon on 30 December. A brief removal of pollutants occurred in the afternoon of 30 December, followed by a rapid re-accumulation during the night. Around noon on 31 December, the surface  $PM_{2.5}$  concentrations decreased to approximately 100  $\mu g m^{-3}$  and remained at this level until noon on 2 January 2024, after which the pollution entered its dissipation stage, marking the end of the event. According to the temporal variation of PM concentrations, this pollution process can be divided into three aerosol accumulation stages (AASs), as indicated by the red dashed boxes in Fig. 3a, with the first two stages representing the dominant accumulation phases. As shown in Fig. 3b, surface temperature and relative humidity exhibited regular diurnal fluctuations beginning on 22 December and showed an overall increasing trend. However, from noon on 28 December to noon on 29 December, both temperature and humidity remained nearly constant. Starting on 30 December, they resumed their diurnal variations, while humidity showed a gradual decreasing trend. The surface pressure showed an inverse relationship with the aerosol concentrations, reaching its minimum when pollution was most severe. Figure 3c shows that horizontal visibility gradually decreased from the evening of 22 December to noon on 30 December, with the lowest visibility below 1 km. From the evening of 31 December to noon on 2 January 2024, visibility again showed a decreasing trend, reaching a minimum of about 5 km. As the pollution dissipated, horizontal visibility improved rapidly.

Figure 4 illustrates the synoptic circulation pattern at 900 hPa over the study region (101°-115° E, 28°-40° N) during the haze pollution episode. During this period, the Guanzhong Plain was dominated by a weak high-pressure system, characterized by a gentle pressure gradient, stagnant moisture transport, and weak wind variability. These conditions resulted in poor horizontal and vertical ventilation, thereby inhibiting the dispersion of pollutants. The persistent weak synoptic forcing and stable stratification provided favourable meteorological conditions for aerosol accumulation over the region.

# 240 3.2 High-resolution vertical structure of the pollution episode

The evolution of this pollution event was observed by both ground-based meteorological instruments and a lidar system. The lidar provided vertical profiles of aerosols and clouds from 0 to 10 km, as well as temperature and relative humidity from 0 to 4 km, with profiles recorded every 2 minutes (in Fig. 5). Periods of lidar maintenance occurred during 23-24 December, 26 December, 27 December 2023, and 2 January 2025, during which the system was not operating, resulting in no data. In Fig. 5c-e, the white areas in the upper layers represent regions of invalid lidar detection, primarily due to severe low-level haze reducing the lidar penetration capability and thus lowering the signal-to-noise ratio of the Raman signal. Fig. 5a and 5c show the 1064 nm Mie-Rayleigh backscatter signal and the 355 nm backscatter coefficient, respectively, with Fig. 5c also indicating


Figure 3. Surface PM and meteorological parameters. Panels (a)-(c) show (a) PM concentrations, (b) surface meteorological parameters, and (c) visibility. In panel (a), the black and blue dotted curves represent the  $PM_{2.5}$  and  $PM_{10}$  concentrations, respectively. In panel (b), the black, blue dotted, and pink dashed curves represent the temperature, relative humidity, and pressure, respectively.

the top height of the aerosol layer. Fig. 5b presents net and total radiation fluxes observed by ground-based instruments. Fig. 5f-h display radiosonde data collected every 12 hours.

As shown in Fig. 5a, three distinct cloud events occurred in the upper atmosphere during the entire pollution episode, on 22-23 December, 28-29 December, and 31 December 2023-1 January 2024. All three cloud events led to a reduction in surface radiation, and the second and third cloud events were accompanied by virga, which caused a significant increase in low-level relative humidity, corresponding to periods of severe surface pollution. Figure 5c clearly shows that the near-surface backscatter coefficient began to increase gradually on the evening of 22 December, reaching high values during 28-30 December, consistent with variations in surface PM concentrations. A temporary decrease occurred on the afternoon of 30 December, followed by a rapid increase in the near-surface backscatter coefficient. From the evening of 31 December to noon on 2 January, another increasing trend was observed, although the values were much lower than before, and by the afternoon of 2 January, the values had decreased to levels comparable to pre-pollution conditions. We divide the pollution episode into two stages: the pollution development stage (21 December to noon on 30 December) and the late pollution dissipation stage (afternoon of 30 December

**Figure 4.** 900 hPa geopotential height (black contours, gpm), temperature (red contours, °C), and wind vectors at 00:00 UTC on (a) 24 December, (b) 26 December, (c) 28 December, and (d) 30 December 2023. The blue triangle marks the location of Xi'an.

to 3 January 2024). Figure 6 presents the continuous distributions of potential temperature, buoyancy acceleration, and vertical velocity.

# 3.2.1 Pollution development stage



The dominant phase of the pollution development stage occurred from 22 to 28 December. Figure 7 shows the vertical distributions of aerosols and temperature during this period, including profiles in the morning, at noon, in the evening, and daily averages. Over the course of the stage, the vertical distributions of aerosols and temperature underwent persistent changes. At the early stage of pollution, the aerosol vertical structure was primarily characterized by a decreasing profile with height, or alternating patterns of decreasing and well-mixed near-surface layers. The temperature vertical structure exhibited pronounced diurnal variations. During periods of severe pollution, the aerosol vertical structure became well-mixed, and the diurnal variations in both aerosol and temperature distributions were not significant. On the evening of 22 December (in Fig. 7a), the backscatter coefficient below approximately 0.7 km was significantly higher than in the morning, while above 0.7 km it was lower than the morning values. The base heights of the elevated TI layer in the morning, noon, and evening were approximately

**Figure 5.** Spatiotemporal distributions of atmospheric vertical structure observed by lidar and radiosonde: (a) range-square-corrected signal (RSCS) at 1064 nm; (b) radiant exposure; (c) backscatter coefficient at 355 nm; (d) temperature measured by lidar; (e) relative humidity measured by lidar; (f) temperature from radiosonde; (g) relative humidity from radiosonde; (h) wind vectors and wind speed from radiosonde.



**Figure 6.** Spatiotemporal distributions of potential temperature, buoyancy acceleration, and vertical velocity: (a) potential temperature; (b) buoyancy acceleration; (c) vertical velocity.

1.3 km, 1.1 km, and 0.7 km, respectively. Subsequently, a surface-based TI layer with a thickness of about 1 km formed near the ground, strongly inhibiting the vertical diffusion of aerosols. This surface-based TI persisted until 24 December, resulting in a stable and well-mixed aerosol layer below 0.5 km. On 25 December, surface heating due to solar radiation promoted vertical aerosol mixing, but also led to the formation of a dome-like aerosol layer around 1.1 km, producing a dome effect that enhanced the TI at this height and restricted aerosols below approximately 1.0 km. Notably, under heavy pollution conditions, the lower atmosphere did not experience a strong TI, but the temperature gradient below 0.8 km was about 3.5 K km<sup>-1</sup>, indicating a stable atmospheric state. The vertical diffusion of aerosols was further limited by the elevated TI layer above, confining aerosols to below the TI height.

Figure 8 presents the hourly vertical profiles of aerosols and temperature during the early stage of aerosol accumulation. The UABL height exhibited a strong positive correlation with the TI layer and a negative correlation with surface  $PM_{2.5}$  concentrations. On 21 December, during the pollution brewing stage, a pronounced stratification was observed at heights of 2-3 km. From the morning of 21 to 22 December, a significant TI layer occurred between approximately 1.5 and 2.5 km, inhibiting the upward transport of aerosols. However, because the TI layer was relatively elevated, vertical transport below its base remained possible, resulting in a decreasing aerosol profile, with higher concentrations near the surface that decreased with height. The average  $PM_{2.5}$  concentrations near the surface remained below 50  $\mu g m^{-3}$ , indicating good air quality. During


Figure 7. Vertical distributions of aerosols and temperature from 22 to 28 December. Panel (p) shows the daily mean  $PM_{2.5}$  concentrations. Gray solid lines indicate hourly vertical profiles of backscatter coefficient and temperature. Yellow solid lines with upward-pointing triangles, green solid lines with downward-pointing triangles, and red dashed lines correspond to observations at 07:00, 19:00, and 13:00 local time, respectively. Black dotted lines represent daily mean values.

the daytime of 22 December, cloud cover significantly reduced downward radiation fluxes (in Fig. 5b), which directly affected near-surface radiative heating and enhanced the stability of the lower atmosphere, limiting the vertical diffusion of water vapor and aerosols. In addition, the residual layer produced a dome effect, causing pronounced warming above approximately 1 km around noon and strengthening the TI in this layer. The elevated TI induced downward buoyancy acceleration, further reducing the UABL height and confining free vertical convection to below approximately 0.9 km (in Fig. 6). Moreover, from 21 to 22 December, winds below 500 m were predominantly northeasterly at 5-10 m s<sup>-1</sup>. In winter in Xi'an, such northeasterly winds are typically associated with pollution transport, initiating the first aerosol accumulation on the afternoon of 22 December. After sunset, surface radiative cooling led to the formation of a stable layer below approximately 1 km, further enhancing aerosol accumulation near the surface. The vertical structure of near-surface aerosols transitioned from a decreasing profile to a well-mixed profile, in which concentrations remained nearly constant with height. On 23 December, weak and variable winds ( $


TI structure. Surface  $PM_{2.5}$  concentrations continued to increase. Considering the buoyancy conditions during this period, the vertical convection range was substantially compressed. Thus, the subsidence of the TI at the UABL, combined with cloud radiative forcing, jointly constrained the vertical convective scale during the early stage of pollution and was the primary factor leading to the accumulation of aerosols.

Figure 8. The early stage of aerosol accumulation. Panels (a) and (b) show the backscatter coefficient and temperature, respectively. In panel (a), black solid lines represent the backscatter coefficient, and dashed lines indicate the UABL layers, with shaded dashed boxes highlighting the UABL heights. When aerosol stratification occurs, red upward-pointing triangles and black downward-pointing triangles correspond to the upper and lower UABLs, respectively. In panel (b), the blue solid line represents the temperature, while red dashed lines and black solid lines correspond to lidar measurements and radiosonde observations, respectively. Shaded dashed boxes indicate the locations of TI layers. The yellow shaded regions represent the magnitudes of PM<sub>2.5</sub> concentrations.

During the mid-stage of pollution development, the vertical structure of aerosols exhibited a well-mixed profile, although the boundary layer remained relatively high, reaching approximately 1 km. As shown in Fig. 5a, from 25 to 27 December, cloud cover was minimal, yet the total downward radiation flux decreased daily, which can be attributed to the radiative forcing of aerosols. At noon on 24 December, aerosols were primarily concentrated near the surface, potentially favoring the onset of the stove effect. However, the surface-based TI restricted vertical diffusion, resulting in negligible changes in surface  $PM_{2.5}$  concentrations. Starting at 12:00 on 25 December, a pronounced aerosol layering was observed around 1.1 km. This layer persisted for 12 hours with a thickness of 200 m. After sunset, surface radiative cooling led to a decrease in near-surface saturation vapor pressure and an increase in relative humidity, providing favorable conditions for aerosol hygroscopic growth





and accelerating liquid-phase and heterogeneous reactions (Cheng, et al., 2016; Tie, et al., 2017). Consequently, both surface PM<sub>2.5</sub> concentrations and aerosol optical parameters increased significantly during the early hours of 26 December.

From noon on 28 December, cloud cover reduced downward radiation fluxes, allowing aerosols to continue accumulating near the surface. During this cloud episode, the occurrence of virga (in Fig. 5a and 5c) further enhanced aerosol hygroscopic growth, resulting in haze formation and further exacerbating pollution. In the afternoon of 29 December, a brief urban heating effect was observed, which could partially enhance turbulent diffusion. However, the limited duration of solar radiation and relatively weak horizontal advection restricted the removal of aerosols. At night, surface pressure decreased, leading to pronounced convective subsidence. Under conditions of high relative humidity, PM<sub>2.5</sub> concentrations and aerosol optical parameters reached their peak values.

Furthermore, three persistent "capping inversion" layers, resembling a dome, were observed during the pollution development stage, hereafter referred to as dome-type TIs, as indicated by the green dashed regions in Fig. 5d. Comparison with Fig. 7d shows that the daily mean surface  $PM_{2.5}$  concentrations tend to increase as the dome height decreases, and decrease as the dome height increases. Although the dome height on 28 December was relatively high, severe pollution was observed, which can be attributed to the hygroscopic growth of aerosols under conditions of high relative humidity.

The hourly evolution of the aerosol vertical structure, atmospheric temperature, and buoyancy acceleration on 25 December is shown in Fig. 9a, while Fig. 9b presents the variations in heating rate, temperature gradient, and TI depth of the elevated TI layer. Under the combined influence of the elevated TI and weak surface convection, a dome-like stratified structure began to develop at the UABL. As illustrated in Fig. 9b, after 10:00 local standard time (LST), the heating rate around 1.2 km was markedly higher than that in the lower layer, indicating enhanced heating near the UABL, which further strengthened the elevated TI. This process not only suppressed turbulent upward transport but also induced localized aerosol subsidence. Meanwhile, surface warming enhanced buoyancy in the lower atmosphere, promoting turbulent uplift of near-surface aerosols. However, the surface heating rate remained relatively weak, and once distinct stratification formed near the UABL, the heating rate exhibited a decreasing trend. This reduction likely resulted from the stratified aerosol layer attenuating downward radiative fluxes and thereby constraining surface warming. Given the mid-latitude location of the study region, the short duration of solar radiation was insufficient to sustain continuous surface heating. Consequently, the subsidence of the upper aerosol layer became the dominant mechanism driving vertical aerosol mixing.

As shown in Fig. 9b, the temperature gradient and TI depth exhibited nearly opposite variations. Between 12:00 and 15:00 LST, the temperature gradient within the elevated TI increased rapidly from approximately 15 K km<sup>-1</sup> to about 37 K km<sup>-1</sup>, while the TI depth decreased from around 550 m to roughly 240 m. This phenomenon was likely associated with the dome effect induced by aerosol stratification. After 16:00 LST, the temperature gradient decreased markedly, accompanied by a corresponding increase in TI depth, which may have resulted from enhanced vertical mixing of aerosols. During nighttime, variations in both temperature gradient and TI depth were primarily governed by surface longwave radiation and the radiative forcing of aerosols. Therefore, the dome effect played a pivotal role during the aerosol accumulation stage. The negative buoyancy acceleration induced by the TI was the dominant factor suppressing vertical diffusion of aerosols, causing the convective mixing height during the mid-pollution stage to remain confined below approximately 1 km.

**Figure 9.** Vertical structure evolution and atmospheric buoyancy on 25 December. Panel (a) shows the vertical structures of aerosols and temperature, and panel (b) presents the buoyancy acceleration. In panel (a), the black circular line denotes the backscatter coefficient, while the blue dashed line and the shaded areas represent the temperature profile and buoyancy acceleration, respectively. The blue (red) shading indicates negative (positive) buoyancy acceleration, corresponding to downward (upward) buoyant motion. Upward- and downward-pointing triangles are used to indicate the TI layer. In panel (b), the green dotted line shows the variation in temperature gradient, and the blue square-dotted line represents the TI depth.

# 3.2.2 Pollution dissipation stage



Between 11:00 and 14:00 LST on 30 December, a surface-based TI with a thickness of up to 500 m developed, effectively suppressing the vertical diffusion of aerosols. However, during this period, the  $PM_{2.5}$  concentrations exhibited a pronounced decline, likely attributable to the drying of near-surface aerosols. As shown in Fig. 5h, northwesterly winds exceeding 8 m s<sup>-1</sup> prevailed above approximately 1 km during daytime on 30 December, theoretically favoring horizontal dispersion of aerosols. Nevertheless, the limited duration of surface heating led to weak buoyancy in the lower atmosphere, insufficient to transport aerosols upward into the layer of efficient horizontal mixing. Under the combined effects of aerosol subsidence and hygroscopic growth, surface  $PM_{2.5}$  concentrations surged again later that day.








In the early morning of 31 December, surface  $PM_{2.5}$  concentrations decreased markedly, likely due to the strong northeasterly winds below 1 km, with wind speeds exceeding 12 m s<sup>-1</sup>. Although the near-surface  $PM_{2.5}$  concentrations decreased significantly, this was not attributed to a genuine clean-air advection. Instead, the reduction occurred because nighttime emissions were relatively weak (Le et al., 2020), causing the northeasterly winds—typically associated with polluted air masses—to become relatively clean, thereby diluting the surface  $PM_{2.5}$  concentrations. At around 14:00 LST on 31 December, a localized stratification was observed, with a formation mechanism similar to that on 25 December. Consequently, the aerosol removal during this episode remained incomplete. The aerosol removal on 2 January 2024 shared a similar mechanism with the first one but featured stronger north-westerly winds. These winds not only enhanced horizontal dispersion but also promoted vertical mixing of aerosols. After noon, as near-surface aerosols underwent drying, rapid surface warming increased buoyancy, facilitating upward transport of aerosols. Meanwhile, strong northwesterly winds exceeding 10 m s<sup>-1</sup> above approximately 0.5 km efficiently advected the uplifted aerosols downstream. Given that the upwind northwestern region is sparsely populated with low emission intensity and relatively clean air, such winds are often regarded as "clean winds." Consequently, the combined effects of radiative heating and strong northwesterly flow resulted in a more complete aerosol removal during this event.

Figures 10a-b show the hourly evolution of aerosol vertical structure, atmospheric temperature, and buoyancy acceleration, while Fig.s 10c-d present the heating rate. Around 10:00 LST on 30 December, localized aerosol subsidence was observed, likely caused by aerosol radiative forcing affecting the thermal structure and buoyancy of the lower atmosphere. By 13:00 LST, near-surface aerosols were mainly concentrated below 0.3 km, promoting the development of a stove effect. The subsequent vertical aerosol distribution followed this pattern. From 14:00 to 17:00 LST, aerosols below 1 km exhibited well-developed vertical mixing, as indicated by the temperature and backscatter coefficient profiles in Fig. 10a. The vertical evolution before 12:00 LST on 2 January 2024 resembled that on 30 December, with near-surface aerosols gradually drying and showing a downward tendency. When near-surface aerosols accumulated, the lower-layer heating rate increased markedly, accelerating the dissipation of the surface-based TI (Fig. 10c at 16:00 LST). As depicted in Fig. 10b, the near-surface backscatter coefficient decreased at 16:00 LST, while the upper-layer coefficient increased, indicating substantial vertical mixing of aerosols. Furthermore, the potential temperature profiles during the afternoons of 30 December and 2 January showed little variation with height, and in some cases decreased with height, implying an unstable atmosphere conducive to vertical convection. Vertical velocity measurements indicate that during the early and mid-stages of pollution, near-surface vertical velocities were relatively small and dominated by subsidence, whereas upward motion prevailed during the late stage of pollution.

#### 3.2.3 Statistical Characteristics

The observed pollution episode was governed by the combined influence of multiple factors. During the pollution development stage, TIs constrained vertical convection, acting as the primary barrier to aerosol vertical diffusion. Cloud radiative forcing further weakened lower-atmosphere turbulence, thereby constraining convective uplift. Aerosols, in turn, modified the atmospheric thermal structure and buoyancy through radiative interactions involving solar shortwave and surface longwave radiation, leading to complex feedbacks with meteorological conditions. During the pollution dissipation stage, enhanced solar radiation restored atmospheric buoyancy and promoted vertical mixing, while horizontal advection facilitated aerosol removal.

Figure 10. Hourly evolution during the pollution dissipation stage. Panels (a) and (b) show the vertical structures of aerosols and temperature, respectively, while panels (c) and (d) depict buoyancy acceleration. In panels (a) and (b), the black circular line denotes the backscatter coefficient, while the blue dashed line and the shaded areas represent the temperature profile and buoyancy acceleration, respectively. The blue (red) shading indicates negative (positive) buoyancy acceleration, corresponding to downward (upward) buoyant motion. Upward- and downward-pointing triangles are used to indicate the TI layer.

During the pollution dissipation stage, sufficient solar radiation enhanced atmospheric buoyancy, facilitating vertical aerosol mixing, while efficient horizontal advection supported aerosol removal.

Hygroscopic growth further amplified particle size and optical effects, intensifying pollution. Statistical analysis combining near-surface relative humidity and  $PM_{2.5}$  concentrations (in Fig. 11) reveals a strong positive correlation below 200 m, indicating that high humidity exacerbates aerosol accumulation. Additionally, the vertical gradient of relative humidity below 300 m decreases with increasing  $PM_{2.5}$ , likely due to surface-based TIs and weak turbulence that limit vertical moisture transport during pollution episodes.

Figure 12a presents the vertical distributions of the aerosol boundary layer height, the elevated TI, and the surface-based TI. Figures 12b and 12c show the mean vertical temperature gradients of the elevated and surface-based TIs along with the




Figure 11. Relationships between PM<sub>2.5</sub> concentrations and (a) relative humidity, and (b) its vertical gradient.

corresponding  $PM_{2.5}$  concentrations. It is evident that the elevated TI is closely associated with the UABL. Both TIs exhibit distinct diurnal variations in the mean temperature gradient, with the mean gradient of the elevated TI negatively correlated with  $PM_{2.5}$  concentrations, whereas that of the surface-based TI shows a positive correlation. Figure 13 presents the frequency distribution of the mean temperature gradient of the elevated TI and the correlation between the TI-top height and the UABL. The results show that TIs with an average temperature gradient ranging from 5 to 10 K km<sup>-1</sup> occur most frequently (36%), and the TI-top height exhibits a significant positive correlation with the UABL height ( $R^2 \approx 0.57$ ).

The mean temperature gradient of the elevated TI exhibits a piecewise negative correlation with  $PM_{2.5}$  concentrations (in Fig. 14a). During intensified pollution, aerosols were primarily concentrated near the surface, enhancing daytime radiative heating in the lower atmosphere and thereby weakening the elevated TI. The observed piecewise pattern may be associated with hygroscopic growth of aerosols during the mid-pollution stage. In contrast, the mean temperature gradient of the surface-based TI shows a pronounced positive correlation with  $PM_{2.5}$  concentrations (in Fig. 14b).

The diurnal rates of change for  $PM_{2.5}$  concentrations, the temperature gradient of the elevated and surface-based TIs are shown in Fig. 15. The diurnal rate of change represents the temporal rate of variation within a day, where positive values denote an increase and negative values denote a decrease.  $PM_{2.5}$  concentrations showed a slight decrease from 00:00 to 10:00 LST, followed by a marked decline between 12:00 and 15:00 LST, and then increased rapidly after 17:00 LST. The elevated TI typically intensified in the early morning, with the strongest rate of change in temperature gradient occurring around 08:00 LST, likely associated with residual aerosol layers from the preceding night. Around noon, enhanced solar radiation warmed both the surface and the top of the aerosol layer, resulting in relatively minor variations in the elevated TI gradient. As solar radiation persisted, enhanced vertical mixing of aerosols contributed to the weakening or disappearance of the elevated TI. Surface-based TI generally strengthened during the early morning and after sunset. The rate of change in surface  $PM_{2.5}$  was strongly coupled with that of the surface-based temperature inversion (TI), with stronger TIs corresponding to higher  $PM_{2.5}$ 

Figure 12. TI characteristics and aerosol boundary layer distribution. (a) Vertical distribution of the UABL and TIs. The black solid line indicates the UABL. The red shaded area represents the elevated TI layer, with red upward-pointing triangles and blue downward-pointing triangles indicating the top and base heights of the elevated TI, respectively. The blue shaded area represents the surface-based TI layer, with orange upward-pointing triangles and black downward-pointing triangles indicating the top and base heights of the surface-based TI, respectively. (b, c) Black dashed lines represent PM<sub>2.5</sub> concentrations, while red and blue solid lines show the mean vertical temperature gradients of the elevated and surface-based TIs, respectively.

concentrations and weaker TIs to lower ones. Notably, TI peaks tended to precede  $PM_{2.5}$  peaks, reflecting the time lag required for aerosol accumulation. Two distinct minima in  $PM_{2.5}$  concentrations were observed—one around 04:00 LST and another around 14:00 LST. The nighttime minimum was primarily due to reduced anthropogenic emissions, while the midday minimum occurred when solar radiation was strongest and both surface-based and elevated TI weakened, enhancing vertical mixing and resulting in the lowest near-surface aerosol concentrations. In contrast,  $PM_{2.5}$  concentrations peaked from late afternoon to nighttime, primarily due to two factors: the strengthening of surface-based TIs and the reduction in solar radiation. These processes suppressed vertical mixing and facilitated the rapid accumulation of aerosols near the surface.

**Figure 13.** Characteristics of the elevated TI. (a) Frequency of temperature gradient occurrence; (b) Relationship between the UABL and the top height of the elevated TI.

Figure 14. Relationships between  $PM_{2.5}$  concentrations and the temperature gradients of (a) the elevated TI and (b) the surface-based TI.

# 4 Conclusions

Using a self-developed Raman-Mie scattering lidar system from Xi'an University of Technology, the vertical structure of the boundary layer during a severe winter haze event in Xi'an was investigated. A calibrated retrieval algorithm was applied to derive the profiles of temperature, relative humidity, and aerosol, providing high-resolution spatiotemporal observations of the vertical structure during the haze episode. By integrating collocated radiosonde and surface meteorological data, the key meteorological characteristics, influencing factors, and interaction mechanisms governing the formation and evolution of this haze event were analyzed. Lidar observations revealed complex vertical distributions of temperature, relative humidity, and aerosols throughout the pollution episode. The results clearly demonstrated the presence of the aerosol dome effect and hygroscopic growth, both contributing to the intensification of pollution, as well as the mechanisms responsible for its dissipation. The




Figure 15. Diurnal rates of change for (a)  $PM_{2.5}$  concentrations, (b) elevated TI temperature gradient, and (c) surface-based TI temperature gradient.

coexistence of elevated and surface-based TIs was a dominant meteorological feature during the haze event, with TI layers and aerosol stratification occurring simultaneously, forming a multilayered aerosol structure.

During the haze development stage, at least three dome phenomena and three cloud-layer episodes were identified. These features weakened vertical mixing within the boundary layer, thereby promoting and intensifying aerosol accumulation. The surface-based TI exhibited a pronounced diurnal variation, serving as another key factor constraining haze dispersion. Notably, the boundary layer height did not decrease significantly, with a minimum value of 0.9 km. Within the surface-based TI, aerosols displayed a relatively uniform vertical distribution, suggesting that the TI layer effectively limited the variability of the boundary layer height. During daytime, radiative forcing associated with different aerosol vertical distributions primarily manifested as upper-layer and lower-layer heating, altering the vertical temperature gradient. In the absence of strong dynamical processes, rapid aerosol subsidence at night could enhance near-surface warming during the following day, facilitating aerosol vertical mixing. However, the nighttime evolution of the aerosol vertical structure was strongly constrained by surface radiative cooling, which was further modulated by cloud cover and other meteorological factors, rendering the process highly uncertain and complex.

Figure 16 illustrates the main aerosol accumulation and dissipation processes during this pollution episode. Under calm and stable synoptic conditions, for a decreasing-type vertical aerosol profile, when the boundary layer height exceeded 1.2 km, aerosol radiative forcing intensified the elevated TI at the top of the aerosol layer, delaying surface warming. The vertical extent of the TI directly influenced atmospheric buoyancy acceleration, resulting in a dome-like stratified structure near the top of the aerosol layer (1 km), with a thickness of 200 m and a duration of up to 12 hours. As surface radiative heating progressed, buoyancy in the lower atmosphere increased, driving turbulent uplift of near-surface aerosols and promoting vertical mixing.



**Figure 16.** Aerosol accumulation and dissipation during the pollution development. The gray shading indicates the aerosol layer. Orange downward arrows represent downward radiative flux, and green upward arrows indicate the reflected component. The red solid line denotes the temperature profile.

However, the extent of this mixing was limited by the duration of surface heating. Subsidence of aerosols induced by nighttime radiative cooling was the primary mechanism breaking the stratification. When aerosols were concentrated near the surface, aerosol radiative heating further accelerated near-surface warming, enhancing turbulent diffusion. Statistical analysis revealed that  $PM_{2.5}$  concentrations were positively correlated with both near-surface relative humidity and the vertical temperature gradient of the surface-based TI. Furthermore, the boundary layer height exhibited a strong positive correlation with the top height of the TI layer ( $R^2 \approx 0.57$ ). For the TI layer near the boundary layer, the temperature gradient showed a piecewise negative correlation with  $PM_{2.5}$  concentrations. Diurnal variations were also evident for  $PM_{2.5}$  concentrations as well as the temperature gradients of elevated and surface-based TIs. Specifically, the temperature gradient of the elevated TI increased during daytime and decreased at night, consistent with radiative heating at the top of the aerosol layer. In contrast, the temperature gradient of the surface-based TI decreased after sunrise and increased after sunset, reflecting the influence of surface radiative forcing.

This study employed lidar observations to investigate the vertical structure of a severe haze event. However, as the measurements were limited to a single site, it should be noted that haze episodes typically occur over a broader region. The vertical structure of haze may vary due to factors such as local topography and synoptic conditions, potentially giving rise to different phenomena. Comprehensive, large-scale observational networks are therefore required to capture these spatial variations, and such studies will be pursued in future research.

Code and data availability. The data and codes related to this article are available upon request from the corresponding author.

Author contributions. Conceptualization: Qimeng Li & Huige Di; Investigation: Qimeng Li; Methodology: Qimeng Li; Software: Qimeng 470 Li & Ning Chen; Writing — original draft: Qimeng Li & Huige Di; Writing — review & editing: Huige Di & Xiao Cheng & Jiaying Yang; Supervision: Huige Di & Dengxin Hua & Qing Yan; Data collation: Ning Chen & Yun Yuan

Competing interests. The authors declare that they have no conflicts of interest related to this work.

Acknowledgements. We express our gratitude to the Xi'an Meteorology Bureau of Shaanxi Province, Xi'an, Shehong Li, Shuicheng Bai, and Mei Cao for providing the relevant supporting data.

*Financial support.* This research was supported by the National Natural Science Foundation of China (Grant No. 42130612), and Shaanxi Province Natural Science Basic Research Plan (Grant No. 2025JC-YBQN-458, 2025JC-YBQN-453).

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
