# Peer review of "Atmospheric vertical structure variations during severe aerosol pollution events based on lidar observations"

_EGUsphere, 2025_

## Author Comment (AC1)

**Responses and Explanations on new the revision**

(From the author of the manuscript egusphere-2025-5393, Qimeng Li)

22 January 2026

**To Reviewer 1**

We gratefully acknowledge the reviewer for the careful review of our manuscript and for providing detailed and valuable comments. We believe that these comments have significantly contributed to improving the scientific quality and presentation of the manuscript. We have addressed all comments point by point, and the corresponding revisions have been incorporated into the revised version.

1) The authors calculate buoyancy acceleration (Eq. 9) using lidar-derived virtual potential temperature. While they state temperature uncertainties are <1 K and water vapor <0.5 g/kg (Table 1), they do not discuss how these errors propagate into the buoyancy and stability metrics. Given that the study relies heavily on small changes in stability to explain the "dome effect," a formal error propagation analysis is necessary to ensure the observed trends exceed the instrument's noise floor.

**Reply:** Thank you very much for your insightful comments. We strongly agree with your suggestion and have added corresponding error propagation analysis for the calculation of buoyancy acceleration. Based on your suggestions, after careful consideration, we have made the following additions.

**The following text shows the revisions to the description:**
According to the theory of error propagation, the uncertainty of buoyancy acceleration can be expressed as:

$$\delta_B = \sqrt{\left(\frac{\partial B}{\partial \Gamma}\right)^2 \delta_\Gamma{}^2 + \left(\frac{\partial B}{\partial \Gamma}\right)^2 \delta_\theta{}^2} \,, \tag{10}$$

In the formula, $\Gamma = d\theta/dz_0$. The lidar system exhibits a high signal-to-noise ratio within the detection range below 4 km at night (2 km during daytime), and the temperature uncertainty is typically less than 0.5 K. Selecting typical boundary layer conditions with a virtual temperature of approximately 280K and a conservative uncertainty of 0.5 K, when the vertical resolution is 37.5m, the uncertainty of the temperature gradient obtained is approximately 0.019 K m$^{-1}$, and the uncertainty of the buoyancy acceleration is calculated to be approximately 0.025 m s$^{-2}$. This value represents the worst-case scenario assuming completely uncorrelated errors. Lidar temperature errors exhibit strong vertical correlation, and buoyancy is essentially derived from vertical gradients. Therefore, the effective uncertainty in buoyancy is substantially smaller. Although this uncertainty may affect very weak buoyancy signals near neutral conditions, the identification of stable layers and strong inversions is robust.

2) The manuscript frequently employs strong causal language that may not be fully supported by the observational evidence. For instance, the authors state that clouds and virga "drove"

suppressing radiative heating and the rapid increase in PM$_{2.5}$. While the temporal correlation is evident, the study does not sufficiently account for confounding factors such as variations in local primary emissions or regional advection during these specific windows. Furthermore, the discussion of the "stove effect" is somewhat contradictory; it is described as "favoring" pollution alleviation, yet the authors simultaneously conclude that surface-based TIs rendered this effect "negligible". A more rigorous analysis, perhaps involving a mass-balance approach or sensitivity tests, is required to disentangle these competing meteorological and chemical mechanisms.

**Reply:** We appreciate your constructive comments. Following your suggestions, we carefully reviewed the manuscript and revised statements where the supporting evidence was insufficient.

For the virga event observed during the night of 28 December, we examined 48-hour backward air-mass trajectories at multiple altitudes (0.5, 1.5, and 3.0 km). The results indicate that airflows at approximately 0.5 km and 1.5 km were predominantly advected from southern regions toward the observation site, with relatively short transport pathways and limited residence over major upwind pollution source regions. In contrast, air masses at 3.0 km originated mainly from the west to northwest and were less directly coupled to the near-surface aerosol layer. These trajectory characteristics suggest that large-scale regional transport was unlikely to be the dominant contributor to near-surface aerosol loading during this period. Meanwhile, strict emission control measures were implemented in Xi'an, including traffic restrictions and industrial emission limits, which likely reduced local primary emissions.

Under these conditions, the observed haze development is more plausibly associated with unfavorable boundary-layer dynamics, particularly the presence of a stable temperature inversion and weak near-surface ventilation, which suppressed vertical mixing and promoted pollutant accumulation. This interpretation is consistent with the concurrent stagnation of low-level air masses and the nighttime radiative cooling environment during the virga event.

**The following text shows the revisions to the description:**

To assess the role of large-scale transport in the development of the pollution episode, 48 h backward air-mass trajectories at multiple altitudes (0.5, 1.5, and 3.0 km) were analyzed for 28 December, which was characterized by severe pollution. The results indicate that airflows at approximately 0.5 km and 1.5 km were predominantly advected from southern regions toward the observation site, with relatively short transport pathways and limited residence time over major upwind pollution source areas. In contrast, air masses at 3.0 km mainly originated from the west to northwest and exhibited weak coupling with the near-surface aerosol layer. Consequently, large-scale regional transport is unlikely to be the dominant factor governing the near-surface aerosol loading during this period. Moreover, strict emission control measures were implemented in Xi'an, including traffic restrictions and constraints on industrial emissions, which may have further reduced local primary emissions.

[Figure]

[Figure]

[Figure]

We apologize for the previous lack of clarity regarding the relationship between the "stove effect" and the surface-based TI, and have made corresponding revisions in the manuscript ("However, the warming of the lower layer must overcome the surface inversion formed by nocturnal radiative cooling, which delays the development of near-surface turbulence and may be the primary reason for the relatively modest changes in surface $PM_{2.5}$ concentrations."). We would like to clarify that, although the stove effect within the lower layer favors warming and may enhance upward turbulent mixing, the surface inversion formed by nocturnal radiative cooling may delay the development of near-surface turbulence.

**The following text shows the partial revisions to the description:**

- Surface-based TI exhibited a clear diurnal variation, with TI peaks observed to precede aerosol peaks. The results suggest that strong elevated TI and weak turbulence in the lower layer may facilitate aerosol accumulation. Cloud layers not only suppress radiative heating but may also enhance near-surface humidity through virga processes, which may be conducive to increases in $PM_{2.5}$ concentrations. During the dissipation stage, the rapid breakdown of TI and enhanced solar heating were critical for pollutant removal, while efficient horizontal transport facilitated the complete clearance of aerosols within the boundary layer.

- Thus, the subsidence of the TI at the UABL, combined with cloud radiative forcing, appears to jointly constrain the vertical convective scale during the early stage of pollution and may be a primary factor contributing to aerosol accumulation.
- As shown in Fig. 5a, from 25 to 27 December, cloud cover was minimal, yet the total downward radiation flux decreased daily, which may be associated with the radiative forcing of aerosols.
- Although the dome height on 28 December was relatively high, severe pollution was still observed, suggesting that factors such as aerosol hygroscopic growth under high relative humidity may have contributed.

3) The study places heavy emphasis on the "stove effect" and "surface-based temperature inversions (TIs)," both of which occur in the lowest few hundred meters of the atmosphere. However, most Raman-Mie lidar systems suffer from a "blind zone" or "overlap effect" in the first 200–500 meters. While the authors mention a "geometric overlap factor correction" (referencing Li et al., 2025), they do not show the overlap function or discuss the minimum height at which the temperature and humidity retrievals become stable. What is the full-overlap height of the system? If the overlap correction is significant below 500m, how can the authors ensure that the "stove effect" observations (often very close to the surface) are not artifacts of the correction algorithm?

**Reply:** Thank you very much for the reviewer's insightful comments. As pointed out, the stove effect and surface-based temperature inversions predominantly occur within the lowest few hundred meters of the atmosphere. The complete overlap blind zone of the lidar system developed in this study is approximately 120 m; therefore, the minimum theoretical detection height of the system is about 120 m.

Although inconsistency between the two rotational Raman channels may exist, the relative relationship between the channels is generally stable and primarily determined by the system hardware configuration. For the present system, the height range affected by incomplete overlap between the two rotational Raman channels extends up to approximately 600 m. To address this issue, corresponding correction procedures have been applied to ensure reliable temperature retrievals within the incomplete-overlap region. This correction approach has been previously published in Acta Optica Sinica (DOI: 10.3788/AOS241641, Li, Q., Di, H., Chen, N., Cheng, X., Yang, J., Bai, S., Dou, J., Yan, Q., Li, S., Xin, W., Wang, Y., and Hua, D.: Detection and correction techniques of atmospheric temperature profiles within the boundary layer during haze days, Acta Op. Sin., 45(3): 0312003, https://doi.org/10.3788/AOS241641, 2025).

We apologize that the technical methodology was not described with sufficient clarity in the original manuscript. In response to the reviewer's comments, we have now provided a more detailed explanation of the relevant techniques and corrections. Specifically, a description of the system pure blind-zone height has been added ("The complete overlap blind zone of the system is approximately 120 m"), Fig. 1 has been revised, and additional details regarding near-surface atmospheric temperature measurements have been included. The detailed descriptions can be found in Response 7.

[Figure]

**Figure 1.** Atmospheric temperature correction. (a) Measured (blue solid line) and theoretical (black dash–dotted line) rotational Raman ratios. (b) Overlap-related quantities, including the measured ratio (black dashed line) and the overlap function (red solid line). (c) Range-square-corrected signals from the elastic (Mie–Rayleigh; thick red solid line), nitrogen vibrational Raman (thin purple solid line), high- and low-quantum-number rotational Raman (black dash–dotted and blue dashed lines) channels. (d) Backscatter ratio (blue dash–dotted line) and rotational Raman ratios (Measured ratio, black solid line; theoretical ratio, black dash–dotted line). (e) Linear regression analysis. (f) BC. (g) Temperature profiles derived from lidar measurements and radiosonde observations, where the black dash–dotted line denotes the radiosonde temperature and the red solid and blue dashed lines represent the corrected and uncorrected lidar temperature profiles, respectively. Shaded areas indicate the corresponding uncertainties.

4) Does the system provide depolarization measurements? If so, these should be included to confirm the presence of virga and characterize the aerosol type. If not, the authors must clarify how they distinguish between high-extinction "heavy haze" and "cloud base" or "virga" using only backscatter and Raman signals, as these features can look very similar in elastic channels.

**Reply:** We sincerely thank you for your valuable suggestion. The depolarization ratio is indeed an important parameter for distinguishing cloud base, virga, and haze, and it also provides useful information for aerosol type characterization. Our lidar system is capable of measuring the depolarization ratio. However, due to instrumental and data availability limitations, part of the depolarization channel data during the study period was missing, and therefore these results were not included in the original manuscript. Following your suggestion, we processed the available depolarization ratio data for the virga event on 1 January 2024. The results show that within the virga region, the depolarization ratio decreases significantly with increasing range-square-corrected signal (RSCS). In addition, the depolarization ratio is markedly reduced above the cloud-base boundary, indicating that the cloud base exhibits typical liquid-water cloud characteristics. In this study, virga identification is primarily based on the combined Mie–Rayleigh backscatter signals at 1064 nm and 355 nm. It is acknowledged that distinguishing the transition region between haze and cloud using a single vertical profile may involve certain uncertainties. However, continuous lidar observations provide highly correlated features in both the temporal and vertical dimensions, which facilitates a more reliable identification of heavy haze, cloud layers, and virga structures. In addition, to further validate this conclusion, we also retrieved corresponding millimeter-wave radar data for the same periods. As shown in the figure, the vertical velocity and echo reflectivity provide additional evidence supporting the occurrence of the virga events.

[Figure]

Similar virga phenomena have also been reported in previous lidar-based studies. For example, Yi et al. (Yi, Y., Yi, F., Liu, F., Zhang, Y., Yu, C., and He, Y.: Microphysical process of precipitating hydrometeors from warm-front mid-level stratiform clouds revealed by ground-based lidar observations, Atmos. Chem. Phys., 21, 17649–17664, https://doi.org/10.5194/acp-21-17649-2021, 2021.) revealed the microphysical processes of precipitating hydrometeors in warm-front mid-level stratiform clouds using ground-based lidar observations and documented comparable virga-like structures during the precipitation evolution, , as shown in the corresponding figure.

[Figure]

**Figure 2.** Time–height contour plots (1 min and 30 m resolution) of the **(a)** range-corrected signal $X$, **(b)** volume depolarization ratio $\delta_v$ measured by a 355 nm polarization lidar and **(c)** water vapor mixing ratio $q_v$ measured by a water vapor Raman lidar on 26–28 December 2017, which exhibited the passage of a warm front and the resulting hours-long light rain. A sliding average of 60 min was applied to the Raman lidar data. The precipitation streaks surrounded by magenta lines are zoomed in to show their details. Shown on the top of the figure are the corresponding photographs of the sky taken by a ground-based camera at our lidar site, with the third photograph exhibiting the sky illuminated by a 532 nm laser beam during the onset of rainfall.

**The following text shows the revisions to the description:**

As shown in Fig. 5a, three distinct cloud events occurred in the upper atmosphere during the pollution episode, specifically on 22–23 December 2023, 28–29 December 2023, and 31 December 2023–1 January 2024. All three events were observed to coincide with reductions in surface radiation. The second and third cloud events were accompanied by virga, during which increased near-surface relative humidity was observed, temporally coinciding with periods of elevated surface pollution. The continuous vertical profile of the lidar Mie–Rayleigh backscatter signal served as the primary basis for the preliminary identification of cloud layers and virga, which could be further corroborated by depolarization ratio measurements or radial velocity observations from millimeter-wave radar (Yi et al., 2021; Zou et al., 2024; Jimenez et al., 2025).

5) The paper identifies three specific "dome-type TIs". The criteria for classifying an inversion as "dome-type" versus a standard elevated inversion should be more explicitly defined. Is this based purely on the geometric shape of the PM5 stratification, or on a specific threshold of radiative heating/cooling rates?

**Reply:** We thank you for your valuable suggestion. The "dome-type TIs" mentioned in the manuscript is primarily defined based on the geometric structure of the observed temperature distribution and the $PM_{2.5}$ stratification. Strictly speaking, it can be classified as a persistent elevated inversion with a dome-like structure, which notably suppresses turbulent motions in the boundary layer. Following your suggestion, we have added a description in the revised manuscript clarifying the basis for defining the dome-shaped inversion.

**The following text shows the revisions to the description:**

Furthermore, three persistent "capping inversion" layers with a dome-like structure were observed during the pollution development stage and are hereafter referred to as dome-type TIs, as indicated by the green dashed regions in Fig. 5d. The identification of dome-type TIs relies primarily on the geometric structure of the vertical temperature profiles and the corresponding aerosol-layer stratification. A comparison with Fig. 7p shows that the daily mean surface PM2.5 concentrations tend to increase as the dome height decreases and decrease as the dome height increases. Although the dome height on 28 December was relatively high, severe pollution was still observed, suggesting that factors such as aerosol hygroscopic growth under high relative humidity may have contributed.

6) To truly claim an "Aerosol-Radiation-Boundary Layer" feedback, could the authors provide a simple estimation of the heating rate induced by the aerosol layer? This would support the "dome effect" hypothesis by showing that the aerosol-induced warming at the top of the layer is sufficient to maintain the observed temperature inversion.

**Reply:** We thank the reviewer for this constructive comment. We have added a simple estimation of the aerosol-induced heating rate in the revised manuscript. The heating rate is diagnostically estimated using a simplified shortwave heating formulation based on the vertical gradient of the net radiative flux. The results indicate that the heating rate is noticeably

enhanced within the aerosol-stratified region, especially near the top of the layer. Although this estimation is simplified, the enhanced aerosol-induced warming is consistent with the presence and persistence of the observed temperature inversion.

**The following text shows the revisions to the description:**

[Figure]

**Figure 9.** Vertical structure evolution and atmospheric buoyancy on 25 December. Panel (a) shows the vertical structures of aerosols and temperature, panel (b) presents the buoyancy acceleration, and panel (c) shows the aerosol heating rate. In panel (a), the black circular line denotes the backscatter coefficient, while the blue dashed line and the shaded areas represent the temperature profile and buoyancy acceleration, respectively. The blue (red) shading indicates negative (positive) buoyancy acceleration, corresponding to downward (upward) buoyant motion. Upward- and downward-pointing triangles are used to indicate the TI layer. In panel (b), the green dotted line shows the variation in temperature gradient, and the blue square-dotted line represents the TI depth.

The hourly evolution of the aerosol vertical structure, atmospheric temperature, and buoyancy acceleration on 25 December is shown in Fig. 9a. Figure 9b presents the variations in temperature tendency, temperature gradient, and TI depth of the elevated TI layer. Figure 9c shows the aerosol heating rate, which is approximately estimated using a simplified shortwave heating formulation in the vertical direction ($Q = 1/(\rho c_p) \cdot dF/dz$), where $\rho$ and $c_p$ denote the air density and the specific heat capacity of air, respectively, and $F$ represents the net radiative flux. Under the combined influence of the elevated TI and weak surface convection, a dome-like stratified structure began to develop near the UABL. As illustrated in Fig. 9b, after 10:00 local standard time (LST), the temperature tendency around 1.2 km increases significantly and is markedly higher than that in the lower layers, whereas the aerosol heating rate in Fig. 9c remains relatively weak. Backward air-mass trajectory analysis shows that the airflow at approximately 1.5 km is advected from southern regions toward the observation site, indicating

a northward transport of warm air. Consequently, horizontal advection may play an important role in the pronounced warming near the upper boundary layer (UABL). Under the combined influence of advective warming and aerosol radiative heating, the elevated TI appears to be further intensified. This process is likely to suppress the upward transport of turbulence and induce local aerosol subsidence. Meanwhile, surface warming may enhance buoyancy in the lower atmosphere, promoting turbulent uplift of near-surface aerosols. However, the heating rate near the surface remains relatively weak. As a pronounced stratification develops near the UABL, the heating rate within the aerosol-stratified layer increases substantially, while the warming rate in the lower layer tends to decrease. This reduction may be partly attributed to the attenuation of downward radiative fluxes by the stratified aerosol layer, which could constrain surface warming. Given the mid-latitude location of the study region, the limited duration of daytime solar radiation may have been insufficient to sustain continuous surface heating. Consequently, subsidence within the upper aerosol layer likely became the dominant mechanism regulating vertical aerosol mixing during this period. These interpretations are based on observational consistency and are subject to uncertainties associated with the simplified heating rate estimation.

7) The Raman ratio correction and geometric overlap factor correction are central to the paper's novelty. While the authors refer to Li et al. (2025) for details, a brief but more comprehensive summary of how the "theoretical rotational Raman ratio" is derived from radiosondes and applied to the "haze layer" retrievals would improve the manuscript's readability and transparency.

**Reply:** Thank you for your comment. In response, we have added a detailed description of the atmospheric temperature correction procedure and revised Fig. 1 and its caption to clarify the retrieval steps and the physical meaning of each panel.

Specifically, the core of the atmospheric temperature correction method is to establish a linear functional relationship between the backscatter ratio and the elastic scattering crosstalk ratio, which is then used to correct the rotational Raman ratio. A high-altitude region unaffected by the geometric overlap factor is first selected and combined with radiosonde temperature profiles for system calibration. Under clear-sky and dry near-surface conditions, the theoretical rotational Raman ratio is derived from radiosonde data and compared with the measured ratio to obtain the geometric overlap factor correction. Subsequently, under strong elastic scattering conditions, the elastic scattering crosstalk ratio is determined using the measured and theoretical Raman ratios. A linear regression analysis between the backscatter ratio and the elastic scattering crosstalk ratio is then performed to derive the system calibration constant, which is finally applied to correct the rotational Raman ratio and retrieve the true atmospheric temperature profile in the elastic scattering region.

In addition, Fig. 1 and its caption have been carefully revised to explicitly describe each panel, including the measured and theoretical Raman ratios, the overlap-related quantities, the range-squared-corrected signals of different channels, the backscatter ratio, the regression analysis, and the resulting temperature profiles with and without correction. These revisions are intended to improve the clarity and reproducibility of the retrieval algorithm. Detailed

descriptions of the algorithm and procedures are provided in Li et al. (2025, DOI: 10.3788/AOS241641).

**The following text shows the revisions to the description:**

The core of the atmospheric temperature correction technique involves constructing a linear functional relationship between the backscatter ratio and the elastic scattering crosstalk ratio, and using the backscatter ratio to correct the rotational Raman ratio. First, a high-altitude detection region that is not affected by the geometric overlap factor is selected and combined with radiosonde temperature profiles to perform system calibration for temperature retrieval. Then, under clear-sky and dry near-surface conditions, the theoretical rotational Raman ratio is derived from radiosonde data (solid blue line in Fig. 1a). By comparing it with the measured Raman ratio (black dashed line in Fig. 1a), the geometric overlap factor (solid red line in Fig. 1b) is obtained, thereby achieving geometric overlap factor correction. Subsequently, the theoretical Raman ratio under strong elastic scattering conditions is derived from radiosonde data (black dash–dotted line in Fig. 1d), and the measured Raman ratio (solid black line in Fig. 1d) is used to calculate the elastic scattering crosstalk ratio. A linear regression analysis of the backscatter ratio and the elastic scattering crosstalk ratio is then performed to derive the corresponding system calibration constant (Fig. 1e). Finally, using this calibration constant together with the measured backscatter ratio (solid black line in Fig. 1f), the rotational Raman ratio is corrected, allowing retrieval of the true atmospheric temperature profile within the elastic scattering region. Figure 1 presents the temperature profiles obtained using the above retrieval algorithm. Detailed descriptions of the algorithm and procedures can be found in Li et al. (2025).

[Figure]

**Figure 1.** Atmospheric temperature correction. (a) Measured (blue solid line) and theoretical (black dash–dotted line) rotational Raman ratios. (b) Overlap-related quantities, including the measured ratio (black dashed line) and the overlap function (red solid line). (c) Range-square-corrected signals from the elastic (Mie–Rayleigh; thick red solid line), nitrogen vibrational Raman (thin purple solid line), high- and low-quantum-number rotational Raman (black dash–dotted and blue dashed lines) channels. (d) Backscatter ratio (blue dash–dotted line) and rotational Raman ratios (Measured ratio, black solid line; theoretical ratio, black dash–dotted line). (e) Linear regression analysis. (f) BC. (g) Temperature profiles derived from lidar measurements and radiosonde observations, where the black dash–dotted line denotes the radiosonde temperature and the red solid and blue dashed lines represent the corrected and uncorrected lidar temperature profiles, respectively. Shaded areas indicate the corresponding uncertainties.

8) The study is conducted in Xi'an, located in the Guanzhong Plain, a narrow basin bordered by the Qinling Mountains to the south. Local phenomena like "dome effects" and "stove effects"

are driven by urban-scale (1–5 km) or valley-scale thermodynamics. A 31 km grid cell is far too coarse to "see" the specific vertical air currents or temperature variations created by the interaction between the city's heat and the nearby mountain slopes. The authors explicitly state that they derive vertical velocity from the "vertical pressure tendency provided by ERA5". Vertical velocity is one of the most difficult variables for reanalysis models to get right at a local level. In a complex basin, the actual vertical motion measured by the lidar (which has a resolution of 3.75 m) might be completely different from the average vertical motion of a 31 km x 31 km block in ERA5. Using a coarse, model-averaged vertical velocity to explain fine-scale aerosol stratification observed by a lidar can be misleading. Specifically, how do the authors justify using a 31 km grid-averaged vertical velocity to interpret aerosol stratification changes observed at a local station? A discussion on the representativeness of ERA5 vertical motion for these specific local events is required.

**Reply:** We thank the reviewer for raising this important and fundamental concern. We fully agree that vertical velocity is one of the most uncertain variables in reanalysis products, especially at local scales and in complex terrain such as the Guanzhong Basin. In the revised manuscript, we clarify that ERA5-derived vertical velocity is not used to quantitatively interpret the fine-scale aerosol stratification observed by the lidar. Instead, it is employed solely to characterize the large-scale synoptic background, such as periods of weak subsidence or ascent that provide a favorable or unfavorable environment for boundary-layer development. While ERA5 vertical velocity cannot represent local-scale updrafts or downdrafts within the urban boundary layer, it can still indicate whether the regional atmosphere is under weak large-scale subsidence or ascent. Such background conditions can modulate the persistence and vertical confinement of aerosols, without directly determining their fine vertical structure. We therefore avoid any one-to-one interpretation between ERA5 vertical velocity and the lidar-observed aerosol layers. The lidar observations remain the primary and direct diagnostic of local aerosol stratification and boundary-layer thermodynamic structure.

**The following text shows the revisions to the description:**
"The vertical velocity used in this study was derived from the vertical pressure tendency provided by ERA5. It is noteworthy that ERA5-derived vertical velocity is not used to quantitatively interpret the fine-scale aerosol stratification observed by the lidar. Instead, it is employed solely to characterize the large-scale synoptic background, such as periods of weak subsidence or ascent that provide a favorable or unfavorable environment for boundary-layer development."

9) Are UABL and PBLH herein different? Kindly ensure if "UABL" (Upper Boundary of Aerosol Layer) is defined at its first mention in the main text and used consistently. Currently, the text occasionally switches between discussing "PBLH" and "UABL".

**Reply:** Thank you for pointing out the critical issue, and sorry for this misstatement. The main research object of this article is the upper boundary of aerosol layer (UABL). According to your suggestions, we have replaced the statement about "PBLH" in the article with "the height of UABL" to ensure the consistency of the research object.